# The Haptomonad Stage of *Crithidia acanthocephali* in *Apis mellifera* Hindgut

**DOI:** 10.3390/vetsci9060298

**Published:** 2022-06-16

**Authors:** María Buendía-Abad, Pilar García-Palencia, Luis Miguel de Pablos, Raquel Martín-Hernández, Mariano Higes

**Affiliations:** 1Laboratorio de Patología Apícola, Centro de Investigación Apícola y Agroambiental (CIAPA), IRIAF—Instituto Regional de Investigación y Desarrollo Agroalimentario y Forestal, Consejería de Agricultura de la Junta de Comunidades de Castilla-La Mancha, 19180 Marchamalo, Spain; rmhernandez@jccm.es; 2Departamento de Medicina Veterinaria y Cirugía Animal, Facultad de Veterinaria, Universidad Complutense de Madrid, 28001 Madrid, Spain; palencia@ucm.es; 3Grupo de Bioquímica y Parasitología Molecular CTS-183, Departamento de Parasitología, Universidad de Granada, 18001 Granada, Spain; lpablos@ugr.es; 4Instituto de Recursos Humanos para la Ciencia y la Tecnología (Increcyt-Feder), Fundación Parque Científico y Tecnológico de Castilla-La Mancha, 02001 Albacete, Spain

**Keywords:** trypanosomatid, honey bee, experimental infection, hindgut, microscopy

## Abstract

*Crithidia acanthocephali* is a trypanosomatid species that was initially described in the digestive tract of Hemiptera. However, this parasite was recently detected in honey bee colonies in Spain, raising the question as to whether bees can act as true hosts for this species. To address this issue, worker bees were experimentally infected with choanomastigotes from the early stationary growth phase and after 12 days, their hindgut was extracted for analysis by light microscopy and TEM. Although no cellular lesions were observed in the honey bee’s tissue, trypanosomatids had differentiated and adopted a haptomonad morphology, transforming their flagella into an attachment pad. This structure allows the protozoa to remain attached to the gut walls via hemidesmosomes-such as junctions. The impact of this species on honey bee health, as well as the pathogenic mechanisms involved, remains unknown. Nevertheless, these results suggest that insect trypanosomatids may have a broader range of hosts than initially thought.

## 1. Introduction

Trypanosomatids are a large group of parasitic protozoa that can infect a wide range of organisms, including plants, insects, and vertebrates. Indeed, some of these species can cause important medical and veterinary diseases, such as *Leishmania* and *Trypanosoma*, and consequently, they have been widely studied [1]. While some of these trypanosomatid species can be transmitted by insect vectors (dixenous), the lifecycle of others may be restricted to only one type of organism, known as monoxenous. Despite representing most diversity of the Trypanosomatidae family, these latter species have received less attention [2,3]. Most monoxenous species infect insects, and while the majority do not appear to harm the host’s health, there are some exceptions that include species that infect honey bees and bumble bees [4]. Moreover, it is significant that insect trypanosomatids have also been found to colonize other host groups, including plants, rats, dogs, bats, or even humans [2,5].

The comparative lack of information about insect trypanosomatids has had a strong impact on the establishment of phylogenetic relationships within the family, which in turn has driven constant modifications in their taxonomy and systematics [6]. This is indeed the case of *Crithidia acanthocephali* and *Crithidia flexonema*: The former was described and isolated from the hindgut of the hemipteran *Acanthocephala femorata* in 1961 [7], while *C. flexonema* was described in 1960 in water strider *Aquarius remiges* (formerly known as *Gerris remiges*) [8]. Following the “one host-one parasite” paradigm that governed the classification of trypanosomatids for many years, these were both considered as different species. However, a recent taxonomic revision based on DNA barcoding [9] showed the sequences of both these species to be identical, leading to the proposal to unify the nomenclature under the name of the first chronological described species, *C. flexonema*. However, to the best of our knowledge there has been no consensus on this proposal and most of the bibliography consulted refers to *C. acanthocephali*. Indeed, the cultures obtained from ATCC are named after this species; thus, from now on this name will be used here to refer to both these species.

*Crithidia acanthocephali* is quite widespread geographically [9], and as previous works have proved, it can proliferate inside insects from different orders, such as Diptera, Coleoptera, or Orthoptera, increasing their mortality [10,11]. Although to date there is little information available regarding this trypanosomatid in honey bees, a recent study detected *C. acanthocephali* for the first time in honey bee colonies in the center of Spain via Ion PGM sequencing, along with other species commonly found in honey bee colonies (e.g., *Lotmaria passim* and *Crithidia mellificae*) [12]. Other trypanosomatid species also detected in this study (such as *Crithidia bombi* or *Crithidia expoeki*) are commonly detected in other hymenopteran hosts such as bumble bees, but they are not usually found in honey bee colonies.

Both *L. passim* and *C. mellificae* have been recently found to modify their promastigote and choanomastigote morphology into an haptomonad form, remodeling their flagella into an attachment pad that allows them to remain attached to the gut walls of their host and cover the epithelial cells [13]. This haptomonad stage and its morphogenesis have been described in other trypanosomatid species, such as *Leishmania*, and it is regarded as an influential factor for parasite survival and transmission [14,15,16]. Those are key features for monoxenous parasites, but acquisition is also crucial for considering an insect as a true host. In this regard, the presence of *C. acanthocephali* in honey bees implies that honey bees acquire this parasite naturally, but how this happens, as well as what occurs inside this host, is still unknown. Here, we report that under experimental conditions, *C. acanthocephali* can establish, thrive, and differentiate into the haptomonad morphotype inside a honey bee’s gut.

## 2. Materials and Methods

The *C. acanthocephali* reference strain (ATCC 30251, American Type Culture Collection) was cultured in vitro to generate the inoculum, establishing serial cultures to infect honey bees with trypanosomatids at the same developmental stage on consecutive days. Starting from an initial concentration of 10^5^ cells/mL, the cells were cultured as described previously for *C. mellificae* and *L. passim* [13], maintaining them at 27 °C in 25 cm^2^ flasks (Corning, New York, NY, USA) in Brain Heart Infusion broth (BHI; Sigma-Aldrich, Merck KGaA, Darmstadt, Germany) supplemented with 10% heat-inactivated fetal bovine serum (HIFBS, Gibco, Thermo-Fisher Scientific, Waltham, MA, USA) and 1% penicillin/streptomycin (Lonza, Basel, Switzerland) [13]. After 96–168 h, the cultures had reached the early stationary phase, and the choanomastigote forms of *C. acanthocephali* were obtained to be used as an inoculum [13]. The cells were counted in a Neubauer chamber and the inoculum concentration was adjusted with Phosphate Buffered Saline (PBS) to 5 × 10^4^ cells/µL.

Brood frames from experimental and control honey bee colonies were kept in the laboratory at 34 ± 1 °C to randomly cage the workers upon their emergence in two experimental groups, infected and non-infected control bees, each with 3 cages of 10 workers per cage (N = 30 workers/group). The bees were maintained for two days at 27 °C in separate incubators (Memmert^®^ IPP500, 0.1 °C, Memmert GmH + Co.KG, Schwabach, Germany) to avoid cross-contamination, and they were fed with 50% sucrose syrup + 2% Promotor L (Laboratorios Calier SA, Barcelona, Spain), which was renewed daily as described elsewhere [13].

To stimulate their appetite, two-day-old bees were starved for two hours. Each bee was then manually inoculated orally with 2 µL of either PBS or the inoculum [17], the latter resulting in a final dose of 10^5^ cells per bee. Worker bees were inoculated twice daily for 12 consecutive days (daily dose/bee: 2 × 10^5^ cells), each dose separated by 6 h, to ensure obtaining images of the trypanosomatids inside the bee’s hindgut (ileum and rectum) [13]. After the second dose each day, they were fed ad libitum as indicated above.

After the 12th day of infection, the bees were sedated with CO_2_ to extract their digestive tract by pulling from the last abdominal segment [17]. Their gut was washed in PBS and placed on 45 µm cellulose nitrate filters (Sartorius, Gotinga, Germany) to keep them stretched during the fixation process [13]. Half of the bees from each cage and group were fixed for 24 h in buffered formalin (10%: Merck KGaA, Darmstadt, Germany) for light microscopy and then they were embedded in paraffin. The microtome sections obtained (4 µm: Leica^®^ 2155, Leica Biosystems, Wetzlar, Germany) were stained with Haematoxylin-Eosin (H&E) [13]. The remaining guts were processed for TEM analysis: first fixing them with Karnovsky fixative at 4 °C to be later stained with 1% osmium tetroxide (Sigma-Aldrich) and dehydrated in a graded acetone series (Panreac Química S.L.U., © ITW Reagents Division, Castellar del Vallès, Spain) and finally embedding them in a graded Spurr resin-acetone series (Sigma-Aldrich). The ileum and rectum were separated and placed in different resin blocks, obtaining semi-thin sections to locate the areas of interest (0.5 µm: Reichert-Jung Ultracut E microtome, Leica microsystems, Wetzlar, Germany^®^), which were then trimmed to obtain ultra-thin sections (60 nm). After performing dual-contrast with 2% uranyl acetate (Thermo-Fisher Scientific) in water and lead citrate Reynolds solution (Merck), the sections were analyzed and photographed (Jeol 1010 and Jeol JEM-1400 Electron Microscope, Tokyo, Japan). Further details of the fixation process can be found elsewhere [13].

## 3. Results

### 3.1. Light Microscopy Analysis

*Crithidia acanthocephali* had colonized the hindguts of infected bees 12 days after infection, whereas no trypanosomatid forms were observed in the control bees. These trypanosomatids were observed to cover the surface of the digestive tract in all the infected bees analyzed, and images were obtained from both the ileum and rectum (Figure 1). The trypanosomatid cells were observed to form clusters and to also organize as monolayers. While the former clusters were more often observed in the ileum, in the rectum, the monolayer arrangement seemed to predominate. No histological changes were observed in the host’s epithelial cells, which suggests that the trypanosomatids cause no evident damage to the intestinal cells.

### 3.2. Transmission Electron Microscopy

Although samples from both the ileum and rectum were processed for TEM, we could only obtain images of *C. acanthocephali* from the latter (Figure 2 and Figure 3). Trypanosomatid cells were found lining the digestive tract surface, and their morphology differed from the choanomastigotes observed in cultures. Instead, they adopted a haptomonad-like form, with their flagella remodeled into an attachment pad. This modification allowed trypanosomatids to attach to the epithelial cells and remain inside the host’s hindgut. Moreover, in accordance with the light microscopy images, these protozoa did not appear to damage host cells.

The magnification of the images allowed us to observe the ultrastructure of the trypanosomatid cells in the magnified images. The nucleus elongated and is centrally located in some cells (Figure 2A) and it moved toward the posterior part of the cell as it approaches the host’s surface (Figure 3A), in which heterochromatin accumulated visibly beneath the membrane (Figure 2A). A prominent disc-shaped kinetoplast could be observed immediately posterior to the start of the flagellum (Figure 2D). A single, large mitochondria can be observed at the peripheral zones of the cells (Figure 2A) and the flagellar pocket seemed to insert up to approximately half the length of the cell body (Figure 2D and Figure 3B,D). This structure (also referred to as the reservoir by some authors [18,19,20]) surrounded the flagellum from its start and throughout its trajectory inside the cell body until it finally emerged at the anterior part of the cell as a modified structure that formed an attachment pad: the flagellopodium (Figure 2A,B and Figure 3B,C,E,F). Doublets of microtubules, not seen in the flagellar pocket, ran along the entire length of the modified flagellum, adopting the typical (9 × 2) + 2 axonemal conformation (Figure 2C,D).

The distal part of this modified flagellum was the point of contact between the trypanosomatid cell and the host’s intestinal surface, forming hemidesmosome-like junction complexes evident as electron-dense areas in the images immediately beneath the flagellopodium membrane (Figure 3B–F). Other junction complexes, such as type A desmosomes, strengthened the entire trypanosomatid cell structure. These desmosomes are responsible for holding the cell body and the flagellopodium together, and they can be seen in the images as dark zones between the membrane of the latter and that of the flagellar pocket (Figure 3C,E,F). To reinforce the entire cell complex, an array of filaments can be seen and they are connected to the axoneme with both types of junctions: hemidesmosome-like complexes and type A desmosomes (Figure 3C,E,F).

Other organelles and cell structures can be observed in the cytoplasm (Figure 2D), including typical trypanosomatid structures such as glycosomes and acidocalcisomes, with different electron densities (Figure 2A). Some vesicles were observed inside the flagellar pocket (Figure 2D) that serve to expand the surface of the reservoir and adapt it to the modifications of the flagellum. A single layer of subpellicular microtubules was found beneath the cell membrane (not shown in the images). In addition, the cells seem to be surrounded by a fiber network of electron-dense particles of unknown nature that seemed to be secreted by the cells themselves (Figure 2 and Figure 3). This fiber network appeared to be more intense at the posterior part of the cell (black arrows). In some cells two axonemes could be observed in two different flagellar pockets (Figure 2C), a clear indicator that events that are part of the multiplication cycle of these haptomonad forms of the organism were underway inside the bee’s rectum.

## 4. Discussion

For the first time, this study describes the presence of the haptomonad morphotype of the *C. acanthocephali* trypanosomatid in the hindgut of *A. mellifera* attached to the intestinal surface through the transformation of the flagellum into an attachment pad. Haptomonad morphology has been observed in the intestinal tract of several insect hosts, such as *Anopheles gambiae* [21], *Phlebotomus papatasi* [22], or *Lutzomyia longipalpis* [23]. This stage was also observed in vitro in culture, since haptomonad cells can attach to synthetic materials [24,25]. Furthermore, a recent study found that *Paratrypanosoma confusum*, a species that infects mosquitoes and that is phylogenetically located between the parasitic trypanosomatids branch and the bodonids (free-living kinetoplastids), had a similar sedentary stage [26]. Thus, the haptomonad morphotype can be considered a common feature of the Trypanosomatidae family.

In terms of honey bees, previous research described both “spheroid” and “flagellated” trypanosomatid forms colonizing the hindgut of worker bees experimentally infected with the species *C. mellificae* or *L. passim* [17,27,28]. A detailed description of the haptomonad form of these two species recently appeared in the honey bee [13], apparently with the aforementioned spheroid morphotype. *Crithidia acanthocephali* adopts a similar morphology, and it was found to colonize both the ileum and rectum. Although trypanosomatids were observed here at both these sites, the latter seems to be the preferred location since it was where the haptomonad forms were observed by TEM. One hypothesis is that the union between the trypanosomatids and the epithelial cells could be thicker in this region than in the ileum such that they might better resist the fixation process when they are in this part of the gut. However, it could also be due to the rectum containing more parasites than the ileum or the nutritional requirements of this trypanosomatid. Sugars and amino acids are thought to be absorbed by rectal cells [29] in what would be an ideal environment for trypanosomatids to grow. The minimal nutritional requirements of these species have been investigated previously by the omission of individual components [30]. It was discovered that this species can use D-ribose as a carbon source, free or as adenosine, although many other carbohydrates enhanced its growth, especially glucose, fructose, and sucrose. These are precisely the major components of the bees’ diet; therefore, they are commonly found in the honey bee’s gut [31]. Another interesting factor is that purine was found to be vital for this species to grow. According to the metabolomics analysis of different honey bee gut regions [31], the highest concentrations of purines can be found in the rectum, followed by the ileum, whereas in the midgut they are barely found at all. Moreover, the cuticle layer that covers the ileum and the rectum seems to play an important role in trypanosomatid adhesion and the formation of junctions [21,28].

Despite the valuable information about the development of this trypanosomatid within the honey bee that is gained by detecting the haptomonad stage, its pathogenic implications remain unclear, as this is also the case with *C. mellificae, L. passim* and other insect trypanosomatids [4]. With the information obtained here, we can only hypothesize about both the pathogenicity of this morphotype and its possible mechanisms of virulence. Based on the lack of histological changes in the gut epithelial cells, it seems most probable that these protozoa are active in the lumen. Indeed, their disposition covering the host surface could hinder nutrient absorption [21,32] and the presence of the uncharacterized secreted particles observed could be implicated in this effect. Thus, future research on the relevance of establishing haptomonads, the biochemical mechanisms implicated, and studies into mortality would be of great interest to determine the pathogenic mechanisms induced by this trypanosomatid species. Nevertheless, remaining attached to the host’s intestinal walls allows trypanosomatids to maintain infection of the host for a long time. Although not much is known about the mechanisms of transmission of monoxenous trypanosomatids, some species of *Blastocrithidia* and *Leptomonas,* among others, form resistant cells or “cysts” that allow their survival under adverse conditions [33,34]. As far as we know, *C. acanthocephali* does not form these resistant forms, so lengthening their stay inside the host could increase the chances of transmission to another individual [35]. However, the establishment and attachment of this trypanosomatid will probably be influenced by other factors. For example, the microbiota present in the honey bee’s gut has been proposed as likely to have a protective role against microorganisms [36].

Based on the “one parasite-one host” paradigm, new trypanosomatid species have traditionally been named according to the host in which they were first described [3]. The molecular characterization of their associations and the specificity of several trypanosomatid species in different heteropteran hosts has proved that these interactions may not be that stringent [35], suggesting more promiscuous host–parasite relationships than were initially thought for monoxenous trypanosomatids. Several authors have used *C. acanthocephali* experimental infection (rectal or haematocele injections) to test this host-parasite specificity in what a priori were considered to be foreign hosts [10,11]. In all cases, dense populations of trypanosomatids were observed to colonize the gut and hemolymph of the host insect, increasing host mortality. Trypanosomatid infection provoked a phagocytic response and the formation of nodules, in which motile flagellates could be observed, sometimes even in stages of division [11], which provides evidence that *C. acanthocephali* can multiply in foreign hosts. Here, events characteristic of trypanosomatid division were detected in the rectum of the honey bees, such as the presence of two flagella on the same cell, suggesting that this trypanosomatid species can truly infect this insect host.

*Crithidia acanthocephali* was recently found for the first time in honey bee colonies in Spain [12] and in bumble bees [37]. It was detected in honey bee colonies throughout the year, at all seasons, and no differences were found between interior and forager bees, indicating it is a common organism in bee colonies. Together with the apparent absence of haplotype differentiation in *L. passim*, *C. mellificae*, or *C. bombi* between their hosts [38], these data suggest that insect trypanosomatids may infect a wider range of species than was previously thought.

## 5. Conclusions

By obtaining the first microscopy images of *C. acanthocephali* in the honey bee hindgut, this trypanosomatid species can apparently adopt the haptomonad stage in this host in order to remain attached to honey bee hindgut cells. The impact of the presence of trypanosomatids and more specifically of *C. acanthocephali* on honey bee health remains unclear, making this an interesting area for further research. Nevertheless, the data presented here suggest that insect trypanosomatids have the potential to infect and multiply in several insect species, which is evidence that these organisms have less strict parasite–host specificity than previously thought.

## Figures and Tables

**Figure 1 vetsci-09-00298-f001:**
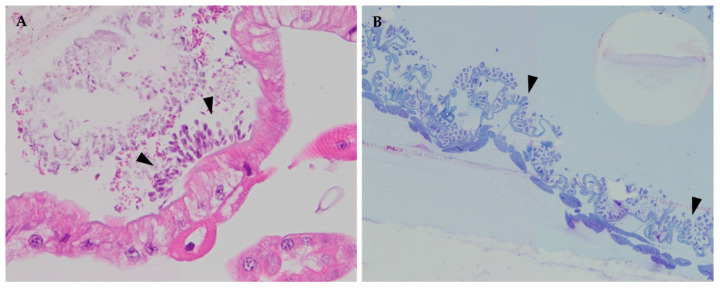
Light microscopy images of the ileum (**A**) and rectum (**B**) of workers infected with *C. acanthocephali*: (**A**) longitudinal section (40×) of the ileum stained with hematoxylin and eosin (H&E), in which trypanosomatid clusters could be observed (arrowheads); (**B**) methylene blue-stained longitudinal semi-thin section (20×) of the rectum, covered by a layer of trypanosomatid cells (arrowheads).

**Figure 2 vetsci-09-00298-f002:**
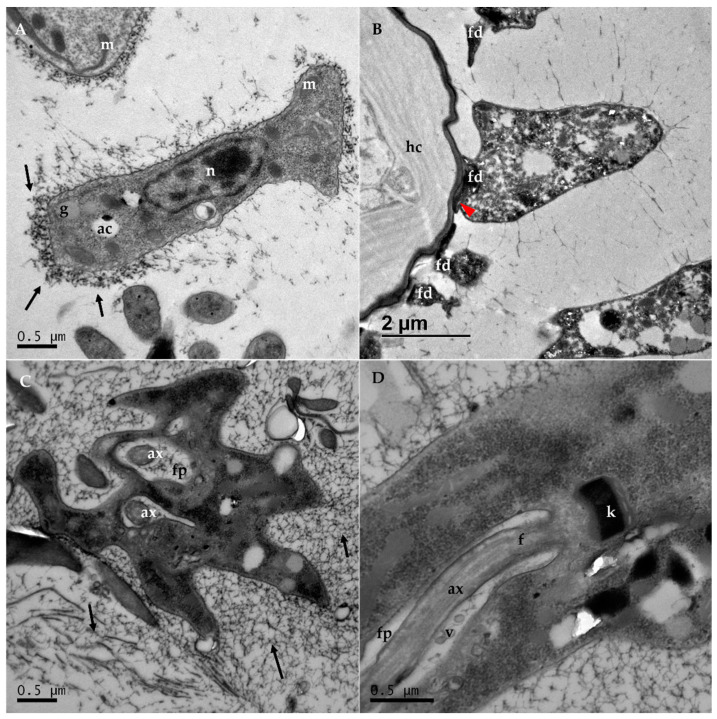
Transmission electron microscopy (TEM) images obtained from the rectum of *C. acanthocephali* infected bees: (**A**) longitudinal section of an adherent haptomonad form, with the elongated nucleus (n) visible on the central part of the cell body, with condensed heterochromatin both centrally and around the nuclear membrane, as well as some organelles as acidocalcisomes (ac) and glycosomes (g). A network of fibers of unknown nature can be observed around the trypanosomatid, especially in the posterior part of the cell (black arrows). (**B**) Haptomonad cell attached to the host cell (hc). Some flagellapodia (fd) that remain attached could also be observed, even though their cell bodies seem to be lost or are not in the same section. (**C**) Cross-section of a haptomonad cell undergoing division, in which two flagella axonemes (ax) can be observed inside both flagellar pockets. (**D**) Detailed longitudinal section of a haptomonad cell. The flagellar body (f), with the axoneme visible (ax), is located within half of the length of the cell body, right before the kinetoplast (k). Some vesicles (v) are observed inside the flagellar pocket, while other organelles, such as the mitochondria (m), are also observed in the cell body.

**Figure 3 vetsci-09-00298-f003:**
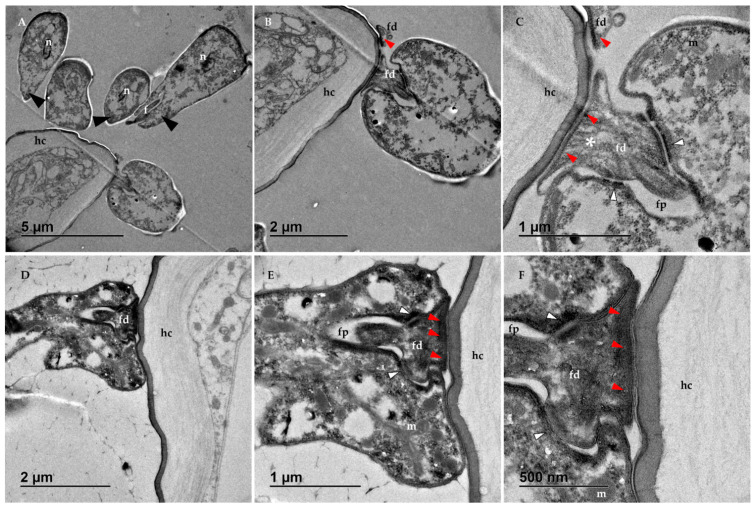
Transmission electron microscopy (TEM) images obtained from the rectum of *C. acanthocephali* infected bees: (**A**) longitudinal section that shows different phases of the attachment process, including haptomonads adhered to the host cell (hc) and non-attached trypanosomatids with their flagella (f) oriented to the host surface (black arrowheads). (**B**,**C**) Details of the haptomonad cells and its flagellopodium (fd), which is surrounded by the flagellar pocket (fp) and is held together with the cell body through the type A desmosomes (white arrowheads). Hemidesmosome-like complexes are observed between the flagellopodium and the host surface (electron-dense material underneath the attachment pad: red arrowheads), while an array of filaments reinforce the entire complex (asterisk). Some organelles could be observed inside the trypanosomatids, such as the nucleus (n) and the mitochondria (m). (**D**) Haptomonad attached to the cuticular layer of the honey bee epithelial cells (hc), with the axoneme (ax) visible at the base of the flagellopodium (fd). (**E**,**F**) Details of the flagellopodium (fd), where type A desmosomes (white arrowheads) and the hemidesmosome-like complexes (red arrowheads) could be observed.

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
