# Peer review of "The Haptomonad Stage of Crithidia acanthocephali in Apis mellifera Hindgut"

_vetsci, 2022, doi:10.3390/vetsci9060298_

Round 1

Reviewer 1 Report

The Abstract mentions the concept of “true hosts” (line 19). This is an important concept that requires a little more explanation in the Introduction. In the context of this manuscript a true host for a monoxenous trypanosomatids is one that can acquire, establish and transmit the trypanosomatid to another true host under natural conditions. There is not too much known about these aspects for many monoxenous trypanosomatids. It should be noted that of these three key features (acquisition, establishment and transmission), it is only establishment that is examined in the manuscript.

The occurrence of haptomonad stages of C. acanthocephali is a good indicator of establishment. The definitive evidence of haptomonad stages is the presence of internal hemidesomosomal (HD) plaques at flagellar attachment sites. The light microscopy images (Figure 1) are consistent with haptomonad stages but do not definitely prove their presence, as the HD plaques cannot be seen by light microscopy. Therefore, the data in Figure 2 are critical. There is only one picture of a HD plaque in Fig. 2b, but this is not very convincing even though it is labelled with “hd”. I cannot see the HD plaque clearly in this image. As this is a key point can the authors provide additional e.g. 3-4 images, showing HD plaques in attached parasites. Arrowheads to indicate their location would be helpful.

Lines 95-96. It would be useful to indicate this is inoculation by the oral route without needing to check details in reference 20 (I could not access reference 19).

Line 240. It should be stated that the transmission mechanism for most of the monoxenous trypanosomatids is unknown. I think that “cyst forms” may be mentioned in some of the literature, perhaps the authors can check this and add a comment and reference(s).

Reviewer 2 Report

I have no substantial objection. The paper is well written and properly structured. The figures are excellent. The references are appropriate.

The main aim of this research was to reveal whether honey bees can be true hosts for Crithidia acanthocephalis, a trypanosomatid species that has recently been detected in honeybee colonies in Spain. The authors described the presence of the haptomonad morphotype of the C. acanthocephali trypanosomatid in the hindgut of A. mellifera. However, due to the lack of histological changes in the gut epithelial cells, most probably these protozoa are active in the lumen. No histological changes were observed in the host’s epithelial cells, which suggests that the trypanosomatids cause no evident damage to the intestinal cells.

The topic is original, but not so relevant for the readers of Veterinary Sciences journal (I think it would better fit in some other journal from MDPI palette (e.g.  ’Biology’, ’Microorganisms’ ...).

In fact, the findings of this study are really new and original, but these protozoa did not appear to damage the host cells. Consequently, I think the paper should be transferred to another journal (related to the biology, not veterinary sciences).

Minor objection:

The statement in line 46 (that Crithidia acanthocephali was first described) and in line 53 (that first described species is C. flexonema) are conradictory, i.e. it is not clead what species was first described.

Reviewer 3 Report

There are not significant comment for authors. The most comment are associated to form some compound words. For instance, "honeybee" or "bumblebees". For correct spelling, please see Kirk, 2021.  https://doi.org/10.1080/0005772X.2021.1982315

I believe that this manuscript is a significant study that updates the importance of trypanosomes in honey bees. To a large extent, we still do not know the importance of this group in the honey bee live and other bees. 

For specific comments, please see pdf file. 

Round 2

Reviewer 1 Report

No further changes required.